

# A nociceptor-specific RNAi screen in *Drosophila* larvae identifies RNA-binding proteins that regulate thermal nociception

Amber Dyson[1], Gita Gajjar[2], Katherine C. Hoffman[1], Dakota Lewis[1], Sara Palega[1], Erik Rangel Silva[1], James Auwn[1] and Andrew Bellemer[1]

[1] Department of Biology, Appalachian State University, Boone, North Carolina, United States
[2] Department of Biochemistry and Molecular Biology, East Carolina University, Greenville, North Carolina, United States

Corresponding author
Andrew Bellemer,
bellemerac@appstate.edu

## ABSTRACT

Nociception is the process by which sensory neurons detect and encode potentially harmful environmental stimuli to generate behavioral responses. Nociceptor neurons exhibit plasticity in which their sensitivity to noxious stimuli and subsequent ability to drive behavior may be altered by environmental conditions, injury, infection, and inflammation. In some cases, nociceptor sensitization requires regulated changes in gene expression, and recent studies have indicated roles for post-transcriptional mechanisms in regulating these changes as an aspect of nociceptor plasticity. The larvae of *Drosophila melanogaster* have been developed as a powerful model for studying mechanisms of nociception, nociceptor plasticity, and nociceptor development. Diverse RNA-binding proteins regulate the development and morphology of larval nociceptors, implying important roles for post-transcriptional regulation of gene expression in these neurons, but the importance of these mechanisms for nociceptive behavior has not been investigated systematically. In this study, we conducted a nociceptor-specific RNAi screen of 112 candidate RNA-binding protein genes to identify those that are required for normal sensitivity to noxious thermal stimuli. The screen and subsequent validation experiments identified nine candidate genes (*eIF2α, eIF4A, eIF4AIII, eIF4G2, mbl, SC35, snf, Larp4B* and *CG10445*) that produce defects in nociceptive response latency when knocked down in larval nociceptors. Some of the genes identified have well-understood roles in the regulation of translation initiation and regulation of nociceptor sensitization in vertebrate and invertebrate animal models, suggesting an evolutionarily conserved role for these mechanisms in regulating nociceptor sensitivity. Other screen isolates have previously described roles in regulating nociceptor morphology and mRNA processing, but less clear roles in regulating nociceptor function. Further studies will be necessary to identify the mechanisms by which the identified RNA-binding proteins regulate sensory neuron function and the identities of the mRNAs that they target.

## INTRODUCTION

Nociception is the process by which the nervous system detects and encodes noxious sensory stimuli to generate behavioral and physiological responses. A common feature of nociception across genera is plasticity of sensory neuron responses to environmental stimuli. For example, nociceptors may become sensitized and increase their sensitivity to noxious stimuli following injury or inflammation. This process of nociceptor sensitization is observed in vertebrates, arthropods, cephalopods, nematodes, and mollusks (*Illich & Walters, 1997*; *Babcock, Landry & Galko, 2009*; *Basbaum et al., 2009*; *Harris et al., 2010*; *Crook, Hanlon & Walters, 2013*; *Sneddon, 2018*). In many cases, changes in nociceptor sensitivity depend on regulated changes in gene expression in nociceptors. Experimental evidence increasingly points to important roles of post-transcriptional control of gene products that regulate the sensitivity of nociceptors at the level of RNA processing and protein synthesis (*Geranton et al., 2009*; *Melemedjian et al., 2010*; *Zhao et al., 2010*; *Melemedjian et al., 2013*; *Barragán-Iglesias et al., 2018*).

Regulation of translation initiation is one of several post-transcriptional mechanisms that regulate nociceptor sensitivity (*Khoutorsky & Price, 2018*). The eukaryotic initiation factor 4F (eIF4F) complex assembles at the 5′ cap of mRNA molecules as the activating step in the initiation of cap-dependent translation (*Merrick, 2015*). Assembly of the eIF4F core subunits (eIF4E, eIF4A, and eIF4G) is regulated by multiple factors including phosphorylation of the eIF4E subunit, which increases translation of selected transcripts (*Pyronnet et al., 1999*; *Walsh & Mohr, 2004*). In rodent and *Aplysia* models, preventing eIF4E phosphorylation blocks nociceptors from becoming sensitized following injury (*Moy et al., 2017*; *Mihail et al., 2019*). These changes in nociceptor sensitivity presumably arise from changes in the translation of pro-nociceptive gene products, exemplified by the mRNA for *brain-derived neurotrophic factor* (*bdnf*) (*Moy et al., 2018*). The *bdnf-201* mRNA isoform is upregulated following sensory neuron injury and requires eIF4E phosphorylation to efficiently associate with ribosomes (*Kim, Lee & Cho, 2001*; *Uchida, Matsushita & Ueda, 2013*; *Moy et al., 2018*). Concurrent experiments found that activity-dependent translation of BDNF in the dorsal root ganglia of mice injected with a PAR2 agonist required eIF4E phosphorylation, although it is possible that other *bdnf* mRNA isoforms are responsible for this effect.

An additional translation initiation mechanism known to regulate nociception is the phosphorylation of the α subunit of eukaryotic initiation factor 2 (eIF2). This phosphorylation event occurs during cellular stress responses and leads to a global reduction in protein synthesis, but also enhanced translation of select transcripts (*Harding et al., 2000*; *Donnelly et al., 2013*). In a chronic inflammation mouse model, eIF2α phosphorylation was enhanced in dorsal root ganglia neurons, while transgenic mice lacking normal levels of eIF2α phosphorylation displayed reduced thermal nociception sensitivity as compared to wild-type controls (*Khoutorsky et al., 2016*).

In addition to translation initiation, other modes of post-transcriptional regulation control the sensitivity of nociception. For example, the Poly(A)-binding protein (PABP) is a major regulator of mRNA stability and translation initiation (*Gorgoni & Gray, 2004*).

Inhibition of PABP function with a PABP-sequestering RNA mimic in mice led to a decrease in allodynia produced by nerve growth factor or interleukin-6 as well as a loss of capsaicin-mediated inflammatory pain (*Barragán-Iglesias et al., 2018*). As another example, the CWC22 spliceosomal protein is needed for efficient mRNA splicing, to support alternative splicing, and to enhance the deposition of the exon-junction complex (EJC) on spliced mRNAs *via* its interactions with eIF4AIII (*Alexandrov et al., 2012*; *Steckelberg et al., 2012*; *Song et al., 2023*). In a mouse inflammatory pain model, CWC22 was required for thermal hyperalgesia and mechanical allodynia, while CWC22 overexpression induced nociceptive hypersensitivity in the absence of tissue damage (*Song et al., 2023*). Finally, pseudouridine synthase enzymes are responsible for post-transcriptional RNA modification *via* the isomerization of uridines in the RNA strand (*Hamma & Ferré-D'Amaré, 2006*). In a larval *Drosophila* nociception model, pseudouridine synthase enzymes in the RluA family are required to maintain normal nociceptor sensitivity, as loss of RluA-1 or RluA-2 function causes hypersensitive nociception phenotypes (*Song, Ressl & Tracey, 2020*). Taken together, these studies suggest that multiple modes of post-transcriptional regulation target mRNAs to regulate the sensitivity of nociception. These include translation initiation, mRNA stability/decay, splicing, and RNA modification but may also include other modes of regulation.

Class IV multidendritic (mdIV) neurons are the polymodal nociceptors of *Drosophila* larvae and are required for nociceptive behavioral responses to thermal, mechanical, and UV stimuli (*Hwang et al., 2007*; *Xiang et al., 2010*; *Zhong, Hwang & Tracey, 2010*). Upon stimulation of the mdIV neurons, *Drosophila* larvae execute a series of 360° rolls around their long body axis (*Tracey et al., 2003*). This behavior has been termed nocifensive escape locomotion (NEL) and has been developed as an experimental paradigm for studying the mechanisms of nociception, as manipulations of nociceptor function cause predictable changes in the latency of the NEL responses, allowing manipulations of nociceptor function to be quantitatively evaluated. While the roles of RNA-binding proteins and post-transcriptional regulation in mdIV function are not well understood, their roles in mdIV development and morphological plasticity are numerous and varied (*Ye et al., 2004*; *Olesnicky et al., 2014*; *Rode et al., 2018*). A genome-wide nociceptor-specific RNAi screen identified 88 RNA-binding protein genes whose knockdown resulted in defects in the morphology of nociceptor dendrites, suggesting widespread roles of post-transcriptional regulation in nociceptor morphogenesis (*Olesnicky et al., 2014*). An RNAi screen in the *Drosophila* S2 cell line identified 47 RNA-binding proteins that regulate the splice-isoform abundance of alternatively spliced mRNAs required for normal nociceptor development and function, including the cell-adhesion molecule gene, *Dscam*, and the voltage-gated sodium channel gene, *para* (*Park et al., 2004*). Finally, an analysis of the nociceptor dendrite remodeling during metamorphosis identified the eIF4A helicase as an essential post-transcriptional regulator of the neuronal plasticity gene, *mical* (*Rode et al., 2018*).

The goal of this study is to identify the functional roles of RNA-binding proteins and post-transcriptional regulation mechanisms in regulating *Drosophila* larval nociception. To accomplish this, we have conducted a nociceptor-specific candidate RNAi screen to identify genotypes that show significant defects in thermal nociception. The candidates

identified in this screen were then subjected to additional validation studies to produce a panel of post-transcriptional regulators as candidates for further study as nociception regulatory factors.

## METHODS

### *Drosophila* stocks and husbandry

The nociceptor-specific driver line used in this study (*ppk1.9-GAL4; UAS-dicer2*) was obtained from the laboratory of Dr. Dan Tracey. UAS-RNAi lines were generated by the *Drosophila* Transgenic RNAi Project (TRiP), Vienna *Drosophila* Resource Center (VDRC) RNAi project, and the National Institute of Genetics-Japan (NIG). Lines were obtained from the Bloomington *Drosophila* Stock Center (BDSC), the Vienna *Drosophila* Resource Center, and from the laboratory of Dr. Liz Gavis. A list of all RNAi lines used in RNAi screen is provided in Table S1.

Larvae used for behavioral experiments were raised on a premixed cornmeal-molasses medium (Nutri-Fly M; Genesee Scientific, El Cajon, CA, USA) at 25 °C on a 12:12 h light: dark cycle.

### Thermal nociception assays

Thermal nociception assays were conducted using established protocols (*Tracey et al., 2003*; *Herman, Willits & Bellemer, 2018*). Briefly, third-instar larvae were washed from vials using distilled water. Larvae were placed in glass petri dishes along with ~10 mg of dry baker's yeast to reduce surface tension, and then sufficient water was removed to leave only a thin layer of water across the entire surface of the plate. Larvae were allowed to recover until they resumed peristaltic locomotion and then individually stimulated along the lateral body wall at the midpoint between the anterior and posterior ends using a thermal probe consisting of a soldering iron filed into a ~6 mm chisel shaped tip plugged into a Variac Variable Transformer (Part No. ST3PN1210B) (ISE, Inc., Cleveland, OH, USA) to control probe temperature. In each trial, the large, flat surface of the probe tip was pressed against the larva, while the narrow chisel tip was pressed against the petri dish. Probe temperature was measured before and after each animal tested using an IT-23 or IT-1E thermistor and BAT-12 digital thermometer (Physitemp, Clifton, NJb). Trials in which the probe temperature varied from the desired value by more than 0.5 °C were not included in further analysis. Trials were recorded at 30 frames per second using a digital video camera mounted on a dissecting microscope and then analyzed using Adobe Premier Pro to determine the latency between probe contact with the larvae and the completion of a 360° rotation. Larvae that did not roll within 10 s were scored as having a latency of 11 s. Larval genotypes were anonymized during behavioral trials and scoring. In the screen, approximately 35 animals were tested per RNAi genotype, although a small number of genotypes had as few as 12 or as many 68 trials. During the screen, negative and positive control genotypes (*e.g.*, no RNAi and *para* RNAi) were included in every day of testing. For validation experiments, at least 35 animals were tested for each genotype. For each validation experiment, a similar number of larvae from each genotype were included in each day of testing.

## Statistical analysis

All thermal nociception assays were analyzed using the two-sided permutation t-test to determine *P* values for comparisons between RNAi knockdown groups and transgenic control groups. Effect size was calculated for each comparison by calculating Cliff's delta along with a bootstrap 95% percent confidence interval, as previously described (*Ho et al., 2019*). Venn diagrams were constructed using tools developed by the Bioinformatics & Evolutionary Genomics group at Ghent University (https://bioinformatics.psb.ugent.be/webtools/Venn/).

# RESULTS

## A genetic screen for RNA-binding protein genes that regulate nociceptor function

To identify RNA-binding protein genes that regulate nociceptor sensitivity, we performed a candidate RNAi screen of 112 genes that were selected based on their previously established roles in regulating nociceptor morphology, controlling splice isoform abundance of nociceptor-expressed mRNAs (*i.e., Dscam, para*, and *Adar*), or their predicted roles in translation initiation (*Park et al., 2004*; *Olesnicky et al., 2014*). UAS-RNAi lines targeting each gene were crossed to a nociceptor-specific *ppk-GAL4; UAS-dicer2* driver line, and cross progeny larvae were tested with a 46 °C thermal probe to determine their thermal nociception response latency relative to GAL4-only negative-control genotypes and positive-control larvae with nociceptor specific knockdown of the *para* voltage-gated sodium channel mRNA (Figs. 1A, 1B).

During the screening process, we also compared the nociceptive response latency among the negative-control genotypes, which were produced by crossing lines sharing the genetic backgrounds of the UAS-RNAi lines (VDRC *isow*, VDRC *yw; attP*, TRiP *yv; attP2*, TRiP *yv; attP40*, and NIG *w1118*) with *ppk-GAL4; UAS-dicer2*. Interestingly, we found that the *yv; attP2* negative control larvae showed a significantly shorter response latency than the other four negative-control genotypes, which were all statistically indistinguishable (Fig. 1C). For this reason, UAS-RNAi lines with VDRC, TRiP *yv; attP40*, and NIG *w1118* genetic backgrounds were analyzed together (Fig. 1A), while UAS-RNAi lines with the TRiP *yv; attP2* background were analyzed separately (Fig. 1B).

To identify thermal nociception-defective genotypes from the screen, we calculated a population mean and standard deviation for all of the larvae tested in each group of RNAi lines. RNAi knockdown genotypes with a response latency two standard deviations longer than the population mean were considered to be nociception-defective candidates. Using this approach, seven candidate genes (*eIF2α, eIF4A, eIF4AIII, eIF4G2, mbl, SC35*, and *snf*) were identified as potential nociception regulators.

## Validation of candidate regulators of nociceptor function

In order to validate the candidate nociception genes identified in the RNAi screen, we subjected RNAi knockdown larvae to a round of validation testing featuring increased sample sizes and comparisons to both GAL4-only and UAS-RNAi-only negative controls. Knockdown of *eIF2α* using the TRiP GLC01598 UAS-RNAi transgene and the *ppk-GAL4;*

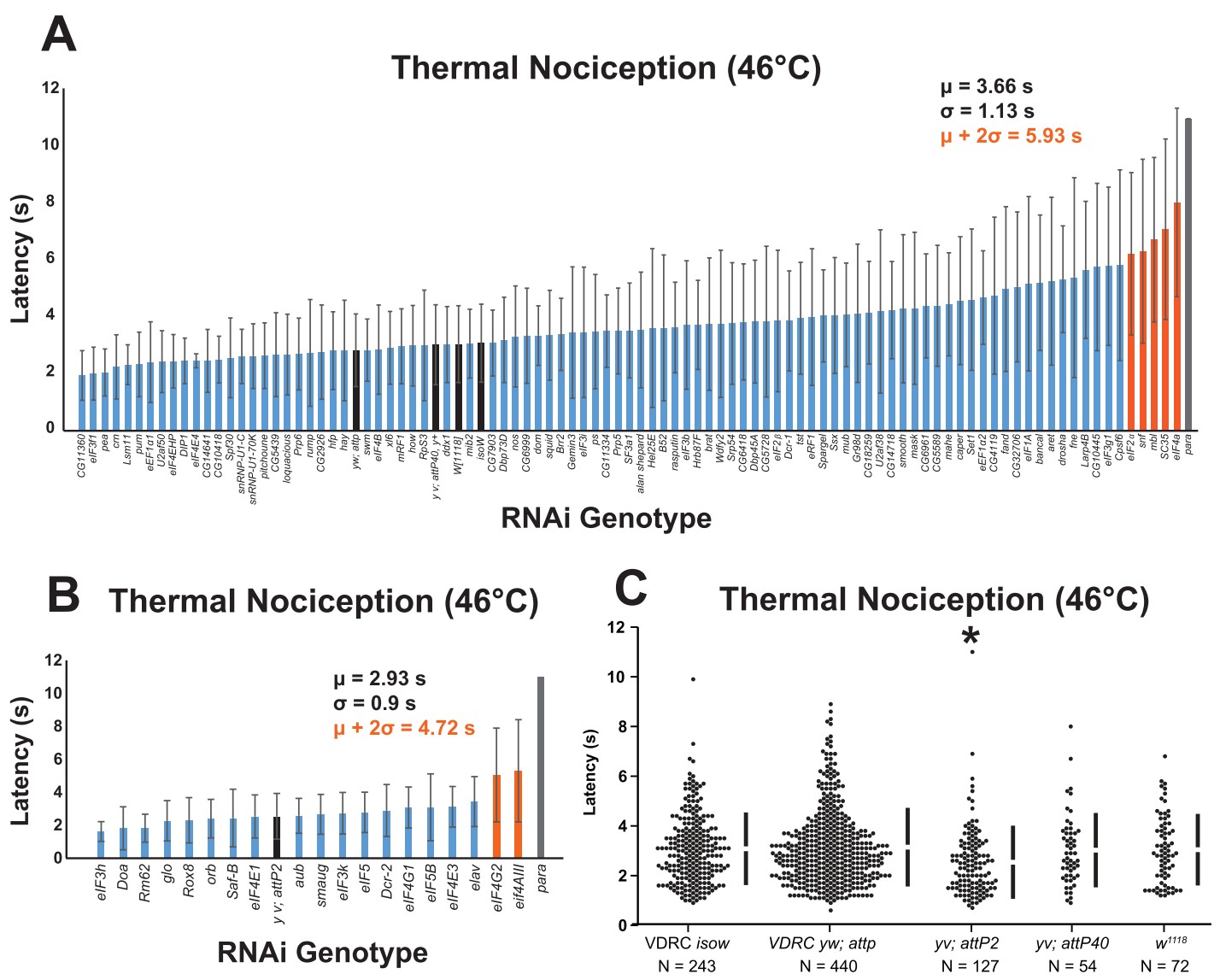

**Figure 1 A nociceptor-specific RNAi screen identifies multiple RNA-binding proteins as candidate regulators of thermal nociception.** (A, B) Candidate RNAi lines from the VDRC RNAi collection, NIG RNAi collection, and the TRiP collection display a wide range of thermal nociception response latencies. Control genotypes are indicated by black bars, while knockdown genotypes are indicated by blue and orange bars. Orange bars indicate knockdown genotypes that showed a mean response latency more than two standard deviations greater than the population mean. Error bars indicate standard deviation ($n \geq 20$ for all groups). (C) The *yv; attP2* control genotype shows a significantly shorter thermal nociception response latency than other control genotypes. The vertical bars to the right of each swarm show the mean latency value (indicated by the gap in each bar) and standard deviation (indicated by the height of each bar around the mean). An asterisk (*) indicates $p \leq 0.0001$ compared to the VDRC *isow* control group by permutation t-test.

*UAS-dicer2* driver line produced a significant longer response latency (5.8 s) than GAL4-only and UAS-only control genotypes (3.1 and 2.8 s) (Fig. 2A). Knockdown of *eIF4A* (VDRC GD14111) also caused larvae to display a significantly longer response latency (5.3 s) than GAL4-only and UAS-only control genotypes (1.8 and 3.6 s) (Fig. 2B). Similar thermal nociception phenotypes were observed for: *eIF4AIII* RNAi (TRiP HMS00442; 6.2 s) compared to its GAL4-only and UAS-only controls (2.7 and 2.9 s) (Fig. 2C), *eIF4G2*
RNAi (TRiP GL00469; 6.0 s) compared to its GAL4-only and UAS-only controls (3.0 and 4.1 s) (Fig. 2D), a second *eIF4G2* RNAi line (TRiP HMS02360; 5.0 s) compared to its GAL4-only and UAS-only controls (3.0 and 4.1 s) (Fig. 2E), *mbl* RNAi (TRiP JF03264; 4.2 s) compared to its GAL4-only and UAS-only controls (2.1 and 2.2 s) (Fig. 2F), *SC35* RNAi (TRiP HMC06150; 4.8 s) compared to its GAL4-only and UAS-only controls (2.8 and 2.7 s) (Fig. 2G), and for *snf* RNAi (TRiP HMC03197; 8.8 s) compared to its GAL4-only and UAS-only controls (3.2 and 3.6 s) (Fig. 2H). The lengthened thermal nociception response latency observed for each of the nociceptor-specific knockdown genotypes tested in these validation studies was consistent with the lengthened response latency observed in the primary screen (Figs. 1A and 1B).

We also tested a small number of RNAi knockdown genotypes (*Cpsf6, Larp4B, CG10445*) for candidates that fell between one and two standard deviations greater than the population latency in order to ensure that a two standard deviation cutoff was not too stringent. Knockdown of *Cpsf6* using the VDRC KK108487 UAS-RNAi transgene and the *ppk-GAL4; UAS-dicer2* driver line produced a thermal nociception response latency (2.1 s) that was significantly different from the latency of the UAS-only genotype (2.7 s), but not significantly different from the latency of the GAL4-only genotype (2.2 s) (Fig. 3A). Knockdown of *CG10445* (TRiP HMC04382; 5.2 s) produced a thermal nociception response latency that was significantly longer than the latencies of GAL4-only and UAS-only controls (2.7 and 2.9 s) (Fig. 3B), and a similar pattern was observed following knockdown of *Larp4B* (TRiP HMC05810; 7.6 s), which caused a significantly lengthened response latency compared to GAL4-only and UAS-only controls (3.6 and 3.5 s) (Fig. 3C). The lengthened thermal nociception response latency for two out of three candidates from this less stringent pool (*CG10445* and *Larp4B*, but not *Cpsf6*) were consistent with the phenotypes observed in the primary screen.

## RNA-binding protein genes have varied roles in nociceptor morphology and function

The RNAi screen for nociception defects and subsequent validation studies ultimately identified nine RNA-binding protein genes (*eIF2α, eIF4A, eIF4AIII, eIF4G2, mbl, SC35, snf, Larp4B*, and *CG10445*) that are required for normal thermal nociception. To understand how this thermal nociception phenotype correlated with previously described morphology and splice isoform-abundance phenotypes (*Park et al., 2004*; *Olesnicky et al., 2014*), we constructed a Venn diagram showing the knockdown phenotypes related to nociception behavior, nociceptor morphology, and splicing of all genes screened (Fig. 4). Two nociception screen isolates (*SC35* and *CG10445*) have been previously shown to produce a splice isoform-abundance phenotype when knocked down. Three screen isolates (*Larp4B, mbl*, and *eIF2α*) have been previously shown to produce a nociceptor morphology defect when knocked down. Two screen isolates (*eIF4A* and *snf*) have been previously shown to produce splicing and morphology phenotypes. Two screen isolates (*eIF4AIII* and *eIF4G2*) have not been identified as producing either splicing or morphology phenotypes, although it should be noted that *eIF4G2* was not tested in the previously described splicing screen (*Park et al., 2004*).

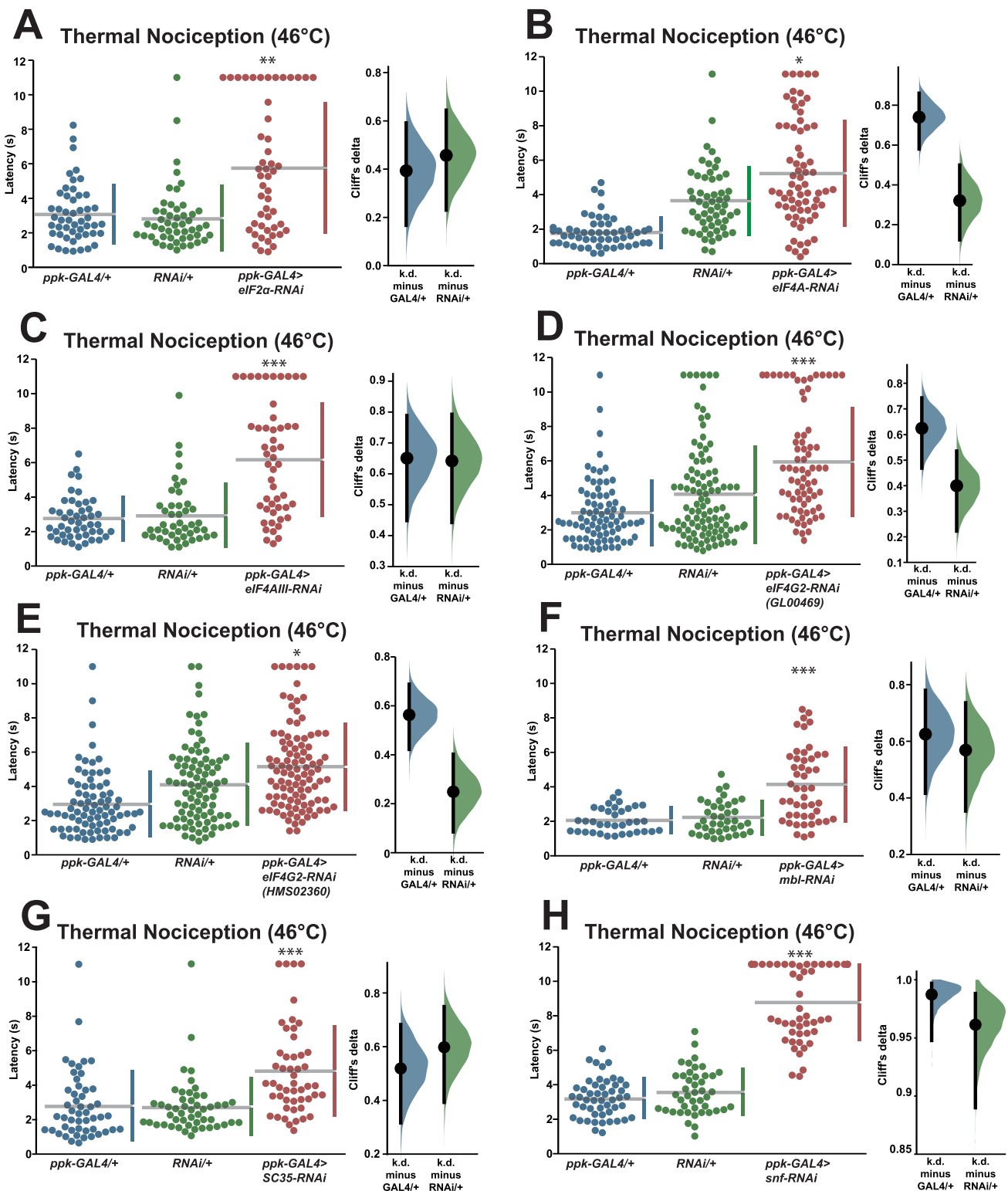

**Figure 2  Validation studies confirm that seven RNA-binding proteins are required for normal thermal nociception sensitivity.** (A–H) Larvae with nociceptor-specific knockdown of 2σ screen candidates show significantly increased thermal nociception latency when compared to controls. In each panel, the left graph displays raw latency data for GAL4-only, UAS-RNAi-only, and RNAi knockdown genotypes. The vertical bars to the right of each swarm show the mean latency value (indicated by the gap in each bar) and standard deviation (indicated by the height of each bar around the

**Figure 2 (continued)**
 mean). The right graph in each panel shows the Cliff's delta effect-size measure for each comparison between the experimental group and control groups. The mean difference is shown as a bootstrap sampling distribution with the mean difference represented by a dot and the 95% confidence interval represented by vertical bars. An asterisk (*) indicates $p \leq 0.0001$ compared to GAL4-only group and $p \leq 0.005$ compared to UAS-RNAi-only group by permutation t-test. Two asterisks (**) indicate $p \leq 0.0005$ compared to GAL4-only group and $p \leq 0.0001$ compared to UAS-RNAi-only group by permutation t-test. Three asterisks (***) indicate $p \leq 0.0001$ compared to GAL4-only group and UAS-RNAi-only group by permutation t-test.

## DISCUSSION

We conducted a candidate RNAi screen for RNA-binding protein genes that produce thermal nociception defectives phenotypes upon nociceptor-specific knockdown. Candidate genes for this screen were selected based on their previously described knockdown phenotypes that included nociceptor morphology defects and changes in splice isoform abundance of nociceptor mRNAs or based on their annotation as translation initiation factors (*Park et al., 2004*; *Olesnicky et al., 2014*). The primary RNAi screen identified seven candidate genes (*eIF2α, eIF4A, eIF4AIII, eIF4G2, mbl, SC35,* and *snf*), and all seven of these candidates, along with two additional genes that reached a less stringent one standard deviation cutoff (*Larp4B* and *CG10445*) produced thermal nociception defects when knocked down in the nociceptors during validation experiments. One candidate that reached the one standard deviation cutoff (*Cpsf6*) did not replicate the thermal nociception defect observed in the primary screen during validation studies. These validation data indicate that the two standard deviation screening criterion was appropriate for avoiding false-positive screen hits, but our validation of some candidates below the cutoff suggests that additional validation experiments could potentially identify additional candidates. Due to the number of genotypes tested in the primary screen and the relatively small number of animals tested for each genotype, it is also likely that the screen data contain some false negative results. As such, the absence of a nociception defect for any RNAi line tested in the screen should not be strongly interpreted. Finally, we note that the knockdown phenotype of one candidate (*eIF4G2*) was further confirmed using a non-overlapping RNAi line. Future studies of each candidate should include phenotypic confirmation with multiple non-overlapping RNAi lines or mutants to confirm the robustness of the nociception phenotype and eliminate the possibility of false positives due to off-target RNAi effects.

For some of the genes that were confirmed to have thermal nociception defective knockdown phenotypes, we can suggest mechanisms of action based on previously published studies. For example, *eIF2α* encodes a component of the ribosomal translation initiation complex that is phosphorylated during cell-stress responses to suppress most translation but boost the translation of specific proteins (*Harding et al., 2000*). Studies of this process in mouse models show that genetic and pharmacological manipulations that reduce eIF2α phosphorylation reduce thermal nociception sensitivity and nociceptor sensitization (*Khoutorsky et al., 2016*). These results suggest that eIF2α phosphorylation may increase the translation of one or more pro-nociceptive gene products or suppress the translation of one or more antinociceptive gene products, and our results showing that *eIF2α* knockdown produces a defect in thermal nociception would suggest a homologous

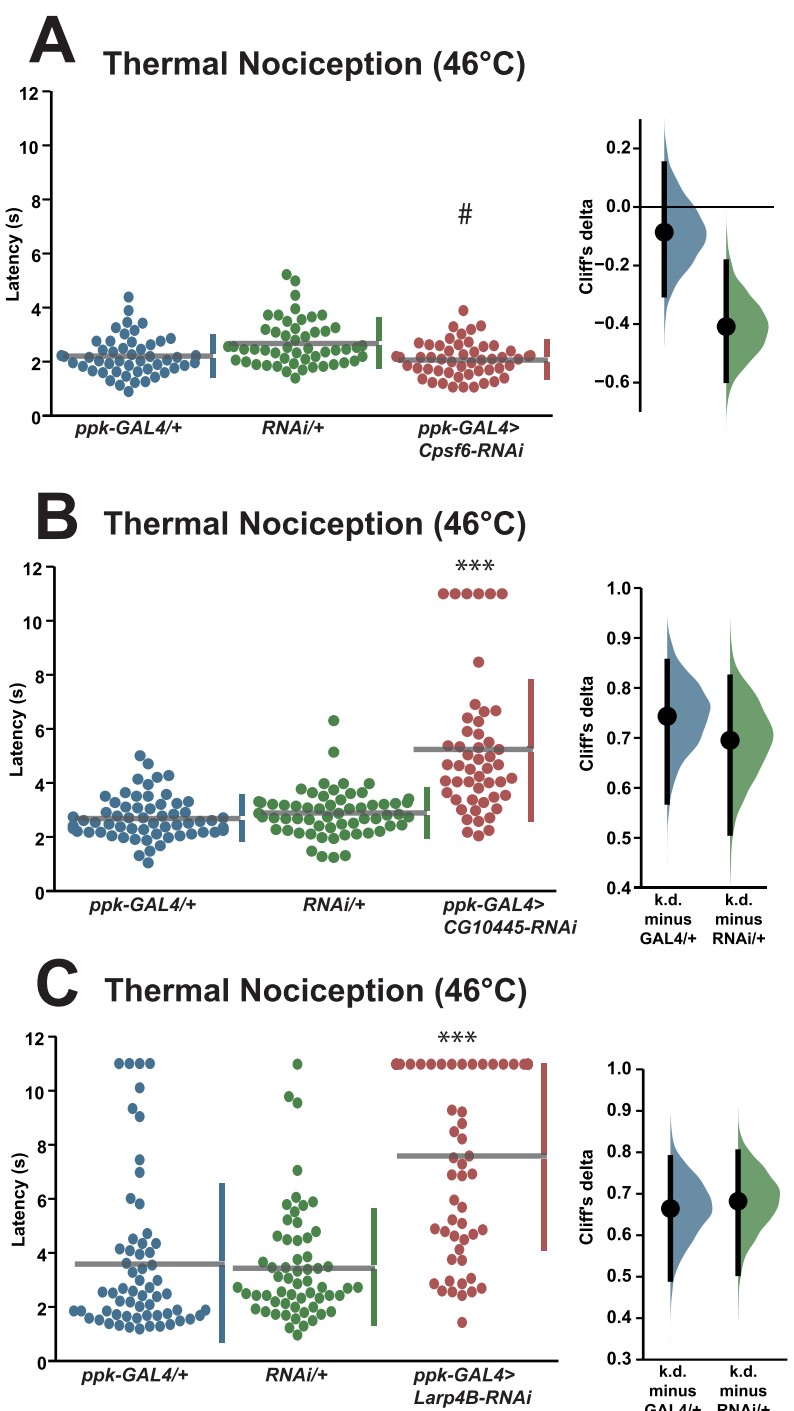

**Figure 3 Validation studies support the screen threshold criterion and identify additional nociception regulators.** (A–C) Larvae with nociceptor-specific knockdown of some 1σ screen candidates show significantly increased thermal nociception latency when compared to controls. In each panel, the left graph displays raw latency data for GAL4-only, UAS-RNAi-only, and RNAi knockdown genotypes. The vertical bars to the right of each swarm show the mean latency value (indicated by the gap in each bar) and standard deviation (indicated by the height of each bar around the mean). The right graph in each panel shows the Cliff's delta effect-size measure for each comparison between the experimental group and control groups. The mean difference is shown as a bootstrap sampling distribution with the mean difference represented by a dot and the 95% confidence interval represented by vertical bars. Three

**Figure 3 (continued)**
asterisks (***) indicate $p \leq 0.0001$ compared to GAL4-only group and UAS-RNAi-only group by permutation t-test. A number sign (#) indicates $p \geq 0.05$ compared to GAL4-only group and $p \leq 0.0001$ compared to UAS-RNAi-only group by permutation t-test.

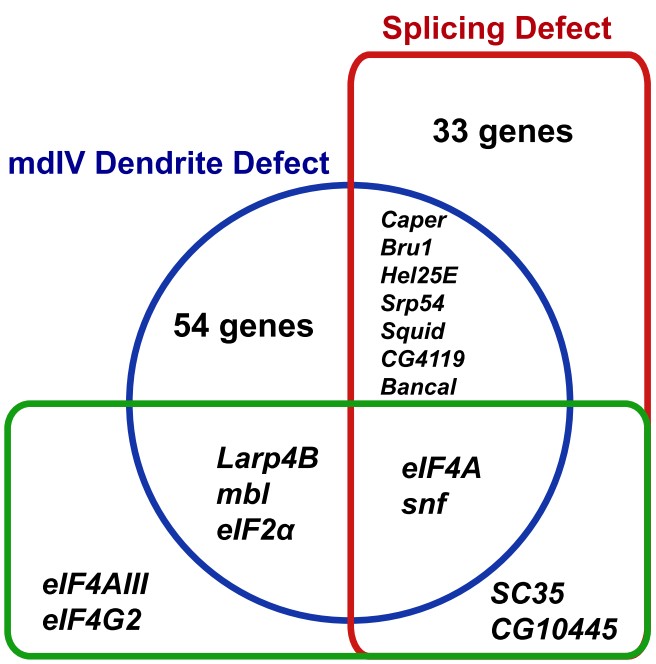

**Figure 4 Identified nociception regulators have diverse roles in regulating nociceptor morphology and nociceptor mRNA splicing.** The Venn diagram shows phenotypes observed for RNAi knockdown of RNA-binding protein genes in this study and in previous studies of mdIV neuron morphology and splice-isoform abundance of neuronal transcripts.

mechanism functions in *Drosophila* nociception. Further studies to investigate eIF2α phosphorylation in larval nociceptors would be warranted to investigate this potential mechanism more fully.

*eIF4A* encodes an RNA helicase that makes up part of the eIF4F translation initiation complex. Previous studies in murine models have demonstrated that eIF4F assembly at the 5′ end of mRNAs is essential for sensitization of nociceptors in multiple experimental paradigms and identified that the mRNA encoding *bdnf* is specifically targeted for increased translation by eIF4F assembly during nociceptor sensitization (*Moy et al., 2017, 2018*; *Mihail et al., 2019*). The sensitivity of *bdnf* to post-transcriptional regulation by eIF4F is likely due to the complexity of the RNA secondary structure found in its 5′ UTR that may be unwound by eIF4A's RNA helicase activity. In *Drosophila* nociceptors, eIF4A function is required for normal nociceptor dendrite branching and for dendrite remodeling during metamorphosis (*Olesnicky et al., 2014*; *Rode et al., 2018*). Our observation that eIF4A knockdown in the nociceptors causes significant defects in the thermal nociception is consistent with a model in which the morphological defects

produced by eIF4A knockdown result in impaired nociceptor function or with a model in which there are pro-nociceptive mRNAs that require the RNA helicase activity of eIF4A for full expression. A role of the eIF4F complex in nociceptor function may also explain the identification of the *eIF4G2* gene in the screen, which encodes a protein with sequence similarity to the mammalian proteins EIF4G1 and EIF4G3 that act as structural subunits of the complex. However, genes encoding other eIF4F subunits (*e.g.*, *eIF4E1, eIF4E3, eIF4G1*) were not identified by the screen, suggesting that there may be some additional complexity in eIF4F assembly and its relationship to thermal nociception in *Drosophila* larvae.

For other genes identified in the screen, direct links to nociceptor function are less clear, but some interesting hypotheses may arise from their nociception defective phenotypes. For instance, *mbl* encodes Muscleblind, a zinc-finger RNA-binding protein that regulates muscle, eye, and neural development (*Begemann et al., 1997*; *Artero et al., 1998*; *Irion, 2012*; *Li & Millard, 2019*). Previous studies have shown that *mbl* knockdown in the nociceptors results in defective dendritic morphology (*Olesnicky et al., 2014*). In photoreceptors and lamina neurons, Muscleblind is required for splicing of mRNAs encoding the cell-adhesion molecule, Dscam2, and for normal dendritic arborization (*Li & Millard, 2019*). This would be consistent with a hypothesis that the *mbl* nociception phenotype may arise from defects in nociceptor development.

*snf* encodes a protein component of the spliceosome that regulates sex-specific alternative splicing of the *sex lethal* mRNA during sex determination (*Albrecht & Salz, 1993*; *Flickinger & Salz, 1994*; *Cline et al., 1999*). Previous reports have indicated that *snf* knockdown produces defects in nociceptor dendrite morphology and *dscam1* splice isoform abundance (*Park et al., 2004*; *Olesnicky et al., 2014*), but further studies are needed to identify the mechanisms by which it affects nociceptor function.

*SC35* encodes a splicing factor of the serine-arginine rich family that regulates alternative splicing (*Fu, 1993*; *Wu & Maniatis, 1993*; *Park et al., 2004*) but has not been previously described as being involved in any nociceptor phenotypes. *eIF4AIII* encodes a protein component of the EJC that is required for regulated subcellular localization of mRNAs and nonsense-mediated decay (*Palacios et al., 2004*). Although specific neuronal roles of eIF4AIII have not been described in *Drosophila*, its role in mRNA localization may have important consequences in highly arborized neurons like the larval nociceptors. Furthermore, genetic manipulations that likely inhibit EJC deposition have been observed to reduce nociceptor sensitivity in vertebrate models (*Song et al., 2023*). *Larp4B* encodes an RNA-binding protein that acts as a negative regulator of organ size and dMyc translation, but is otherwise incompletely characterized (*Funakoshi et al., 2018*). *CG10445* encodes a putative chromatin remodeling complex subunit that regulates the abundance of *para* splice isoforms (*Park et al., 2004*; *Schneiderman et al., 2009*), but is otherwise poorly characterized. Further investigation of each of these screen candidates is needed to understand their potential roles in nociceptor function.

The results of this screen raise some important questions about the mechanisms of RNA-binding protein function in the nociceptors and the relationship between nociceptor morphology and function. Although widespread roles for RNA-binding proteins have been described for regulation of nociceptor morphology in *Drosophila* larvae and for

nociceptor function in murine models, the RNA targets of these proteins and the mechanisms by which they regulate nociceptor form and function are largely unknown, with mRNAs such as *bdnf* serving as an exception. There are, however, examples of nociceptor genes that undergo regulation at the level of translation initiation or alternative splicing. The *para* voltage-gated sodium channel gene is regulated post-transcriptionally to regulate channel abundance and sodium currents in *Drosophila* motor neurons (*Muraro et al., 2008*). It is also required for nociception behavior under the control of larval nociceptors (*Zhong, Hwang & Tracey, 2010*). Thus, it is possible that the translational regulators identified in this screen may share the *para* mRNA as a target to regulate nociceptor function. The *dTrpA1* gene, which encodes a heat-activated ion channel required for nociceptor function, is alternatively spliced to produce isoforms with differing functional properties (*Neely et al., 2011*; *Zhong et al., 2012*; *Gu et al., 2019*). Thus, it is possible that regulated alternative splicing of the *dTrpa1* mRNA by any of the candidates identified in this screen could have important functional consequences for nociceptor sensitivity.

Our results also suggest that the relationship between nociceptor morphology and function is complex. For example, when the *eIF4A* gene is knocked down in the nociceptors, it produces significant defects in dendrite morphology and thermal nociception behavior. This might suggest a causal relationship between these two phenotypes in which morphological defects prevent the neuron from functioning properly. However, other genes (*e.g., RpS3*, *miB2*) were previously characterized as producing severe dendrite morphology defects when knocked down (*Olesnicky et al., 2014*) but were apparently dispensable for thermal nociception behavior in our screen. Furthermore, genes such as *SC35, CG10445, eIF4AIII*, and *eIF4G2* were identified here as producing severe nociception defects upon knockdown, but had not been identified as required for normal nociceptor morphology in previous genome-wide RNAi screens. This suggests that significant defects in nociceptor morphology may not always lead to obvious changes in behavior. Understanding the relationship between morphology and behavior may require more comprehensive studies that examine a wider range of morphological and behavioral parameters.

## CONCLUSIONS

This study used a nociceptor-specific screening approach to identify RNA-binding proteins that are required for normal sensitivity of *Drosophila* larvae to noxious thermal stimuli. Nine RNA-binding protein genes (*eIF2α, eIF4A, eIF4AIII, eIF4G2, mbl, SC35, snf, Larp4B* and *CG10445*) were identified that produced a thermal nociception defective phenotype when knocked down specifically in the nociceptors, and there was strong correspondence between phenotypes observed in the primary screen and subsequent validation studies. Multiple genes identified in the screen are known regulators of translation initiation with previously characterized roles in regulation of nociception (*eIF2α, eIF4A*, and *eIF4G2*), suggesting that translation initiation is an evolutionarily conserved mechanism for regulating nociceptor sensitivity. Additional genes identified in the screen have been previously described for their roles in regulating nociceptor

morphology and mRNA splicing. Further studies are suggested to identify the mechanisms by which post-transcriptional regulation controls nociceptor function.

### Funding
The authors received no funding for this work.

### Competing Interests
The authors declare that they have no competing interests.

### Author Contributions
- Amber Dyson conceived and designed the experiments, performed the experiments, analyzed the data, prepared figures and/or tables, authored or reviewed drafts of the article, and approved the final draft.
- Gita Gajjar performed the experiments, analyzed the data, prepared figures and/or tables, authored or reviewed drafts of the article, and approved the final draft.
- Katherine C. Hoffman performed the experiments, analyzed the data, prepared figures and/or tables, authored or reviewed drafts of the article, and approved the final draft.
- Dakota Lewis performed the experiments, analyzed the data, prepared figures and/or tables, and approved the final draft.
- Sara Palega performed the experiments, analyzed the data, prepared figures and/or tables, and approved the final draft.
- Erik Rangel Silva performed the experiments, analyzed the data, prepared figures and/or tables, and approved the final draft.
- James Auwn analyzed the data, prepared figures and/or tables, authored or reviewed drafts of the article, and approved the final draft.
- Andrew Bellemer conceived and designed the experiments, analyzed the data, prepared figures and/or tables, authored or reviewed drafts of the article, and approved the final draft.

### Data Availability
The raw behavioral data are available in the Supplemental File.

### Supplemental Information
Supplemental information for this article can be found online at http://dx.doi.org/10.7717/peerj.18857#supplemental-information.

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
