# Peer review of "A nociceptor-specific RNAi screen in Drosophila larvae identifies RNA-binding proteins that regulate thermal nociception"

_PeerJ, doi:10.7717/peerj.18857_

## Round 0.1 · original submission · Minor Revisions

As you can see, both reviewers were very positive about the manuscript and think it is worthy of publication. However they also have noted some areas that need addressing prior to acceptance. Please look carefully at these comments, and ensure your address them in a clearly annotated resubmitted version of the manuscript. Please pay particular attention to the notes about clarification of any methodological factors.

Reviewer 1 ·

Basic reporting

Line 153 RNA-binding for many of the factors investigated is indirect. The alpha subunit of eIF2 alpha is not the RNA-binding region. Similarly, the RNA-binding activities of the mid domain of 4G probably aren't physiologically important as they are non-specific. Suggest the author reword.

Line 71 - note that the Bdnf isoform used in the Moy et al study is not the major one expressed in the rodent DRG (Struebing FL et al Differential Expression of Sox11 and Bdnf mRNA Isoforms in the Injured and Regenerating Nervous Systems. Front. Mol. Neurosci., 01 November 2017). Because this impacts the 5'UTR, it is unlikely that the Moy et al data relate to pro-nociceptive cap-dependent translation.

eIF4G2 is a translational repressor. eIF2 alpha is a translational repressor (with exceptions), eIF4A is a translation activator, and EIF4A3 is intimately involved in EJC definition and NMD. These roles are not well explained in the manuscript. It is somewhat odd that the authors focus on cap-dependent translation in the introduction given that most of the hits from the screen are for other processes and that the key genes impact translation in opposing directions (4A and 4G have directly opposing functions). I would suggest replacing the section starting on line 58 with a succinct explanation of what the hits from the screen do - broadly speaking.

Line 250 - The authors imply that EIF2alpha phosphorylation promotes translation of nociceptive genes, but isn't it equally likely that it represses translation of antinociceptive genes? I don't think there is sufficient evidence in support of their suggestion and thus it should be removed or expanded to encompass alternative possibilities.

Line 210 - The authors forgot to include a conclusion from this line of experiments such as - we suspect that the more stringent cutoff was therefor justified or something along those lines

Experimental design

No concerns.

Validity of the findings

Are the results in A or B significant? This seems important. I know it is shown in figure 4 so not a big deal but would be nice to include if known.

Controls are excellent.

Additional comments

I have mild concerns that none of the RNAi is validated but I appreciate the challenge of doing so given the small number of cells that is impacted. I wonder if the authors have considered orthogonal approaches to test key findings. For example, eFT226 is a very potent eIF4A inhibitor (there are others) and happily translation factors are exquisitely conserved. It would strengthen the paper tremendously to provide a different line of evidence that disruption of one or more targets with an approach other than RNAi alleviated thermal hyperalgesia.

Reviewer 2 ·

Basic reporting

A) Figures clear, but...

B) On figures, is it possible to improve the visibility and comparability of the little gaps in the bars? To me, assessing means is just as important as, or more important than seeing the dots representing each response, given the t-test is a test focused on the means.

C) Recommend to avoid interpretation in the Results section (eg line 197, 205), save for Discussion

D) Regarding the Venn diagram... Line 226 Clarify that these isolates were included in both previous studies but had no relevant phenotypes?

E) Improve citation format throughout, eg. use semicolons to separate references

F) Capitalize Aplysia, line 67

G) Inconsistent spelling of Muscleblind, line 276

H) Line 348, change 'suggesting' to 'suggested'?

I) probe tip temperature checking... per animal or per session? Clarify.

J) Can you describe the orientation of chisel tip and location on animal's body.

K) Clarify lines 67-68... eliminate double-negative, or word repetition...

L) Include details of N in Methods.

Experimental design

A) In the validation phase, the addition of the UAS-only control is important, but the lack of an additional non-overlapping RNAi construct means that there is a chance that some apparent positives are false, despite the claim in line 197. This reviewer would not consider these candidates 'confirmed' (eg. Line 243) until such an experiment produced a similar phenotype. This reviewer would consider the manuscript acceptable if even one of the final candidates were to be confirmed to this level of rigor.

B) With only one RNAi line tested per gene, there is a chance that some apparent negatives are false, despite the advantages of having UAS-Dicer2 in the genotype. The authors should point this out clearly, and also make it clear that N ranged from 12 to 65 for experimental genotypes. I would not like to worry that readers might not realize that the initial screen is very preliminary and should be interpreted as such.

C) Do any of these genotypes produce a sufficiently strong hypomotility phenotype as to confound the results? Post-transcriptional processes like translation initiation could be expected to affect other general health biologies. That analysis could be added to the subsequent analysis of these candidates.

D) Not sure why latencies above 10 seconds are calculated as 11 seconds... why not 10? On a related note: how does the ceiling effect of all those non-responders affect the statistics? This reviewer imagines this would have an effect on variance that could affect the results.

E) With an approximately 3 second latency for normal animals, it might be unlikely that any genotypes that would produce hypersensitivity would be identified, due to basement effect. But is it impossible that there are any RNA-binding proteins whose function might be to reduce sensitivity, and would these not be just as important to identify? Maybe a lower temperature, or different probe tip shape would allow better detection of hypersensitivity in addition to hyposensitivity.

F) The N for normal controls is much higher than for experimental genotypes, is this related to operator blinding procedures? Ideally, the operator should be blinded to treatment and without having a normal animal in the assay, it's hard to imagine how they would be blinded. So, was this was included in the design and if so, it should be clearly described in Methods.

G) For snf, might be useful to analyze relative to sex of the larvae? That will require increased N.

Validity of the findings

A) Validity concern addressed above in 2D

B) Data provided and well organized

Additional comments

The authors are to be applauded on the scale of this effort to identify novel regulators of sensitivity in primary nociceptors, and on their stated eventual goal of connecting to membrane electrophysiology mechanisms in this cell. The implications on the pain field are significant. With few exceptions as noted above, the experimental design is logical and the manuscript is clearly and concisely written.

---

## Round 0.2 · accepted · Accept

The authors have addressed the reviewers' comments, and the paper is now acceptable for publication.